# Conditional Entropy: A Potential Digital Marker for Stress

**DOI:** 10.3390/e23030286

**Published:** 2021-02-26

**Authors:** Soheil Keshmiri

**Affiliations:** Advanced Telecommunications Research Institute International (ATR), Kyoto 619-0237, Japan; soheil@atr.jp

**Keywords:** conditional entropy, stress diagnosis, digital marker, EEG

## Abstract

Recent decades have witnessed a substantial progress in the utilization of brain activity for the identification of stress digital markers. In particular, the success of entropic measures for this purpose is very appealing, considering (1) their suitability for capturing both linear and non-linear characteristics of brain activity recordings and (2) their direct association with the brain signal variability. These findings rely on external stimuli to induce the brain stress response. On the other hand, research suggests that the use of different types of experimentally induced psychological and physical stressors could potentially yield differential impacts on the brain response to stress and therefore should be dissociated from more general patterns. The present study takes a step toward addressing this issue by introducing conditional entropy (CE) as a potential electroencephalography (EEG)-based resting-state digital marker of stress. For this purpose, we use the resting-state multi-channel EEG recordings of 20 individuals whose responses to stress-related questionnaires show significantly higher and lower level of stress. Through the application of representational similarity analysis (RSA) and K-nearest-neighbor (KNN) classification, we verify the potential that the use of CE can offer to the solution concept of finding an effective digital marker for stress.

## 1. Introduction

Stress depletes our capacity for reasoning [1,2,3] via strengthening the memories of stressful experiences [4,5,6] and reinforcing the state of fearful arousal that urges the need for rapid defense mechanisms [7,8,9]. With its far reaching impact [10], it is not only a major underlying cause of many emotional disorders [11], but also one of the top leading causes of death in the human population [12]. It is apparent that finding effective diagnostic markers for stress can significantly improve our ability for the intervention and treatment of stress-related mental health problems that affect over 400 million individuals globally [13].

Recent decades have witnessed a substantial progress in the identification of stress markers through the utilization of measurable bodily responses [14,15,16,17,18,19,20,21,22,23]. For instance, Smets et al. [24] used individuals’ demographics, baseline psychological information, and five consecutive days of free-living physiological and contextual measurements to derive a digital phenotype that could predict their high depression, anxiety, and stress scores. In the same vein, Jacobson and colleagues [25] utilized actigraphy data from healthy individuals on the one hand and patients with major depressive and bipolar disorders on the other hand to report a high prediction accuracy (89.0%) on patients’ status. Their results also suggested that actigraphy data may establish reliable measures for predicting changes in patients’ symptoms across a two-week period.

Considering the ample evidence of the negative impact of stress on large-scale brain networks [26,27], the utilization of brain activity presents a unique opportunity for the identification of effective stress markers [28,29]. For example, Zhang et al. [30] used pre- and post-stress functional magnetic resonance imaging (fMRI) scans to devise a data-driven discriminative spatial linear filtering. They reported a high (>75.0%) pre- versus post-stress state prediction accuracy. Their data-driven approach also confirmed the involvement of the default mode network (DMN) and the salient-network (SN) in the brain stress response [31,32,33]. However, the use of fMRI limits the applicability of their approach in more naturalistic settings. The search for effective brain-based stress markers can undoubtedly benefit from the use of EEG, given its relative ease-of-use, portability, and potentials for the study and analysis of brain function [34,35,36,37].

A growing number of EEG-based studies have provided promising results on the utility of this technology for the solution concept of a digital marker for stress. For example, Peng et al. [38] used EEG recordings from human subjects with high versus moderate stress symptoms. They showed that the brain activity of the individuals with high chronic stress was associated with higher left prefrontal power. However, their observations were limited by the fact that they solely focused on EEG recordings from three prefrontal sites (Fp1, Fp2, and Fpz). Minguillon et al. [39] conducted a study based on stress and relaxation periods on six healthy subjects who performed the Montreal Imaging Stress Task (MIST). They observed correlations between prefrontal relative gamma power (RG) and the expected stress level and with the heart rate (HR). They also observed that RG showed a higher discriminative power than alpha asymmetry, theta, alpha, beta, and gamma power in prefrontal cortex (PFC). Using the mental arithmetic (MA) task with three different levels of difficulty, Al-Shargie and colleagues [40] showed that there were significant differences in EEG response between the control and stress conditions. They further utilized multi-class support vector machine (SVM) and reported a high stress level classification accuracy (94.79%). Jebelli et al. [41] proposed a real-time EEG-based stress classification approach based on the online multitask learning (OMTL) paradigm [42]. They reported high stress prediction accuracy on two datasets: a publicly available dataset of EEG and peripheral physiological signals of 32 participants who watched 40 one-minute long excerpts of music videos (DEAP [43], 71.14%) and a dataset from three real construction sites [44] (77.61%).

Entropic measures have also manifested as a robust feature for the study and analysis of the individuals’ mental states [45,46,47,48]. For example, Zheng and Lu [49] showed that the use of entropy for EEG-based emotion classification outperformed such feature spaces as differential asymmetry (DASM), rational asymmetry (RASM), and power spectral density (PSD). García-Martínez et al. [50] used DEAP to compare the utility of sample entropy (SE), quadratic SE (QSE), and distribution entropy (DE) for calm versus distressed mental state discrimination. They found that QSE was able to identify distress with high accuracy (≈70.0%), which was further improved (≈80.0%) using tree-based classification. Martínez-Rodrigo and colleagues [51] applied delayed permutation entropy (DPE) (an extension of amplitude-aware permutation entropy (AAPE) [52]) and permutation min-entropy (PME) [53] on the DEAP dataset for EEG-based distress recognition. Their results indicated that calmness was associated with higher entropy values than distress. García-Martínez et al. [54] studied the use of conditional entropy (CEn) and its correction (CCEn) for the classification of calm versus distress. They reported ≈65.0% discriminative power of CCEn, which was similar and complementary to the previous results based on QSE [50]. The use of CCEn and QSE in their study also suggested a synchronization between opposite frontal and parietal brain regions in both hemispheres in which an increase in distress was associated with an increase in irregular activity in left frontal and right parietal areas.

The overview of the literature on the use of brain activity for the identification of a stress marker shows a substantial progress in this direction. In particular, the use of entropic measures [50,51,54] is very appealing, considering (1) their suitability for capturing both linear and non-linear characteristics of EEG time series [55] and (2) their direct correspondence with brain signal variability [47,56,57,58,59,60,61]. At the same time, these findings rely on external stimuli to induce the brain stress response. In this respect, Van Oort et al. [27] observed that the use of different types of experimentally induced psychological and physical stressors by most previous studies could potentially yield differential impacts on the brain response to stress. They further asserted that such variations must be dissociated from more (potentially) general patterns.

The present study takes a step toward addressing this issue by introducing conditional entropy (CE) as a potential EEG-based resting-state digital marker of stress. For this purpose, we use the resting-state EEG recordings of human subjects from the Max Planck Institute Leipzig Mind-Brain-Body Dataset [62]. The participants in this study did not perform any stress-related task and only completed the Multidimensional Mood State (MDBF) questionnaire [63] prior to their resting-state EEG recordings (five-point Likert scale, from one (not at all) to five (very much) [62]).

The contribution of the present study is twofold. First, we extend the previous research that was based on brain stress response to psychological and physical stressors to the case of resting-state brain activity. In so doing, we show that CE highlights the effect of stress on the brain’s frontoparietal network [64], which plays a pivotal role in self-referential processing [65], emotion [66], and social cognition [67]. Second, we show that CE presents an effective EEG-based resting-state digital marker for quantification of the brain’s distributed response to stress.

The remainder of this article is organized as follows. Section 2 provides details on the dataset, resting-state EEG recordings, and their pre-processing. Section 3 summarizes the analysis steps adapted in the present study. Section 4 presents the results. A discussion along with the limitations and future direction of the present study are presented in Section 5 and Section 6.

## 2. Materials and Methods

### 2.1. Dataset

In the present study, we used the Max Planck Institute Leipzig Mind-Brain-Body Dataset [62]. It is comprised of 227 participants in two age groups: the younger group (153 participants, 45 females, age: mean (M) = 25.1, median (Mdn) = 24.0, standard deviation (SD) = 3.1) and the older group (74 participants, 37 females, age: Mdn = 67.0, M = 67.6, SD = 4.7). The individuals in this dataset underwent a comprehensive set of neurophysiological (from fMRI and EEG to cardiovascular measures, blood samples, urine drug tests, etc.) and psychological tests (including 6 cognitive tests and 21 questionnaires on personality traits and tendencies, eating, addictive and emotional behavior, etc.). The resting-state EEG recordings were acquired from 216 (age: M = 38.61, Mdn = 30.0, SD = 20.14) out of 227 participants. It is this subset of 216 individuals that was used in the present study.

### 2.2. EEG Acquisition

The sixty-two-channel resting-state EEG recordings (61 scalp and a VEOGelectrode below the right eye) were arranged according to the 10–20 extended localization system (also known as the 10–10 system [68]) and referenced to FCz. The EEG signals were bandpass filtered between 0.015 Hz and 1 kHz and further digitized at a 2500 Hz sampling rate. Each EEG session, per participant, was comprised of 16 blocks with two resting-state settings: eyes-closed (EC) and eyes-open (EO). Each EC and EO setting consisted of 8 blocks with a 60 s duration, per block. Every participant completed these two resting-state EEG recordings. For every participant, the EEG recording session started with EC.

### 2.3. EEG Pre-Processing

The raw EEG was first down-sampled from 2500 Hz to 250 Hz and subsequently bandpass filtered within 1–45 Hz using an eight-order Butterworth filter (i.e., four orders in both directions to minimize zero-crossing distortions by low-frequency drifts [69]). The data were then split into EC and EO, each comprised of eight 60 s blocks. For each of these blocks, the EEG channels were visually inspected, and the channels that were affected by such issues as frequent jumps/shifts in voltage and/or poor signal quality were rejected. Furthermore, data intervals that contained extreme peak-to-peak deflections or large bursts of high-frequency activity were also removed (identified through visual inspection). Next, the dimensionality of the data (i.e., the EEGs’ channel-dimension) was reduced by performing principal component analysis (PCA) and keeping PCs (≥30) that explained 95.0% of the variance. This step was then followed by independent component analysis (ICA) on the temporal (i.e., the EEG channels’ data points) dimension of the data using the Infomax (runica) algorithm (step size: $\frac{0.00065}{log(number\;of\;channels)}$, annealing policy: when the weight change > 0.000001, the learning rate was multiplied by 0.98, the stopping criterion maximum number of iterations was 512, or the weight change < 0.000001). Subsequently, components that reflected eye movement, eye blink, or heartbeat-related artifacts were removed. Retained independent components for the EO (M = 19.70, range = 9.0–30.0) and EC (M = 21.40, range = 14.0–28.0) conditions were then back-projected to the sensor space for further analysis. These analyses were performed using EEGLAB [70] (Version 14.1.1b) for MATLAB (Delorme and Makeig, 2004).

In the present study, we used participants’ EC trials. In addition to the above pre-processing steps, we also detrended (using MATLAB 2016a’s inbuilt detrend function) the EEG signals, per participant, per channel, prior to any further computation and analysis.

### 2.4. Choice of Individuals with High and Low Stress Responses

We used individuals’ responses to “neuroticism” from the big-five of personality (also known as the five-factor model (FFM) [71]), “worries” and “tension” from the Perceived Stress Questionnaire (PSQ) [72], and the State-Trait Anxiety Inventory (STAI-G-X2) [73] to systematically identify those whose responses to these questionnaires indicated a significantly higher and lower level of stress.

We found that 40 participants (i.e., out of 216 that were included in the present study; Section 2.1) missed at least one of these responses. Therefore, we discarded these 40 participants. This brought the total number of individuals that were included in our study to 176.

For these participants, we determined the 95.0% confidence interval of their responses to each of FFM’s “neuroticism”, PSQ’s “worries” and “tension”, and STAI-G-X2’s “state-trait anxiety” (Appendix A, Table 1). We then marked those participants, per item, whose responses to all of four items were strictly below the lower bound of the respective item-wise CI (i.e., *p* < 0.025). Similarly, we identified those individuals whose responses were strictly above the upper bound of these items’ CI. We labeled these participants as LOW (14 participants) and HIGH

Further investigation of these groups indicated that there were only one older male (age range, 60–65) in the HIGH and one older male (age range, 70–75) in the LOW groups. Additionally, we also observed that there was only one younger female (age range, 30–35) in the LOW group. Therefore, we decided to discard the older participants and the female participants from our further analyses. This resulted in discarding 4 participants from the LOW group (one older male, one younger female, and two older females). After this step, the total number of participants in the LOW group became 10 (age range: Mdn = 20–25, M = 28–33, SD = 14.38–14.38). In the case of the HIGH group, we discarded seven younger females and one older male, bringing the total number to 18.

We balanced the number of individuals in the HIGH and LOW groups as follows. First, we formed a 1 × 4 questionnaire-response vector (one entry per neuroticism, worries, tension, and STAI trait anxiety) for each participant in the HIGH group. To account for the varying scale among these items, we z-score normalized these vectors and then computed their all-pair Euclidean distances. Last, we chose 10 participants from the HIGH group whose Euclidean distances between their questionnaire-response vectors were smallest (age range: Mdn = 20–25, M = 21–26, SD = 2.11–2.11).

### 2.5. EEG Channels’ Inclusion

The pre-processing steps that were applied on the EEG recordings’ channel dimension (i.e., PCA pre-processing step; Section 2.3) resulted in missing EEG channels in the case of some of the participants. To balance the EEG channels for all participants, we therefore checked for the EEG channels that were common among all 20 individuals in the HIGH and LOW groups. We found (Figure 1) that 53 EEG channels were commonly available in all participants’ pre-processed EEG. These channels were FP2, AF7, AF3, AFZ, AF4, AF8, C5, C3, C1, Cz, C2, C4, C6, CP5, CP3, CP1, CPZ, CP2, CP4, F5, F3, FZ, F1, F2, F4, F6, F8, FT7, FC5, FC3, FC1, FC2, FC4, FC6, FT8, P7, P5, P3, P1, PZ, P2, P4, P6, P8, PO9, PO7, POZ, PO3, PO4, PO8, and O1, OZ, O2.

### 2.6. Conditional Entropy Computation

Let *X* and *Y* represent two EEG channels. The conditional entropy (CE) of *X* given *Y*, represented as *H*(*X|Y*), is computed as [74]:(1)$$\begin{array}{c} H\left(X|Y\right)=\sum_{y\in Y}p\left(x\right)H\left(X|Y=y\right)\\ =-\sum_{y\in Y}\sum_{x\in X}p\left(x,y\right)log\left(p\left(x|y\right)\right) \end{array}$$

CE quantifies the amount of uncertainty that remains about *X* (i.e., its information content) when *Y* is known/observed. Equivalently, the CE of *X* given *Y* can be computed based on the relation between the amount of information that is shared by *X* and *Y* (i.e., their mutual information (*MI*)) and *X*’s entropy [74]:(2)$$\begin{array}{c} MI(X;Y)=H(X)-H(X|Y)\\ \Rightarrow H(X|Y)=H(X)-MI(X;Y) \end{array}$$
where *H*(*X*) and *MI*(*X;Y*) are the *X*’s entropy and the mutual information between *X* and *Y*.

We used Equation (Equation 2) to compute the expected CE for each EEG recording (53 channels), per participant in the HIGH and LOW groups. Specifically, for each EEG channel $c_{i},i=1,\ldots ,53$, we computed its expected CE as:(3)$$CE\left(c_{i}\right)=\frac{1}{N}\sum_{j}H\left(c_{i}|c_{j}\right),\;j=1,\ldots ,N,\;j\neq i$$
where *N* = 53 is the number of EEG channels (Figure 1). Therefore, $CE(c_{i})$ in Equation (Equation 3) quantifies the average information content in EEG channel $c_{i}$ after all its pairwise conditional entropy values with respect to the remaining EEG channels (i.e., all information in $c_{i}$ that could be explained by observing $c_{j},j=1,\ldots N,\;j\neq i$ channels) are accounted for.

## 3. Analysis

First, we performed representational similarity analysis (RSA) [75,76] with Pearson correlation as the similarity distance on the HIGH and LOW groups. For this test, each participant was represented by a 1 × 53 vector whose entries corresponded to the CE values of their EEG recordings. To determine whether the CE vectors could distinguish between the HIGH and LOW groups, we then used these vectors and applied the KNN classifier with one-holdout on them. This resulted in 20 rounds of KNN classification in which every participant was held-out as a test once, and KNN was trained on the remaining 19 participants’ data.

Next, we investigated the contribution of each EEG channel to the classification of the HIGH and LOW groups. For this purpose, we performed permutation importance [77] on the KNN classifier. We then recalculated the RSA and KNN results from the first step using the reduced number of channels based on permutation importance outcome. We used scikit-learn [78] for KNN and permutation importance (i.e., “permutation_importance”). RSA analyses were carried out in MATLAB.

We computed CE values using the Python version of JIDT [79]. With regard to CE computation, there is a crucial point that deserves further explanation. While computing the CE, it is possible to opt for parametric approaches, thereby bypassing the estimation of the joint and conditional probabilities in Equation (Equation 1) and instead utilizing available analytical forms. For instance, assuming *X* and *Y* are Gaussian random variables, then one could make use of $H\left(X\right)=0.5\times \log\left(2\pi e\sigma_{X}^{2}\right)$ [74] (p. 244), $H\left(Y\right)=0.5\times \log\left(2\pi e\sigma_{Y}^{2}\right)$, and $MI\left(X,Y\right)=-0.5\times \log\left(1-\rho^{2}\right)$ [74] (p. 252) (where $\sigma_{X}^{2}$ and $\sigma_{Y}^{2}$ are the *X* and *Y* respective variance and $\rho^{2}$ denotes their correlation) to compute their CE. However, such an approach would inevitably require an assumption about the shape of the underlying probability densities of the random variables under consideration (e.g., the Gaussianity of *X* and *Y* in the above example) [80]. To avoid imposing such a constraint/assumption on our data, we opted for non-parametric estimation of the CE [79]. When using such non-parametric estimators, a crucial step is to verify whether the observed non-zero estimates are due to such issues as limited data and/or non-stationarity that could be present in time series. Therefore, it is important to ensure that the observed non-zero entropy values are indeed significant through the application of the permutation test on the surrogate data during their computation [55]. We achieved this objective through the “null distribution and statistical significance” functionality that is available in JIDT [79] using 100 rounds of permutation tests (computeSignificance(100) in JIDT).

## 4. Results

Figure 2 shows the grand average topographic map of the HIGH and LOW groups’ CE values. The computed CEs for the LOW group highlight the brain’s frontoparietal network. On the other hand, these values appear to be dispersed in the case of the HIGH group.

Pairwise RSA with Pearson correlation similarity distance indicated that (Figure 3A) the HIGH and LOW groups were associated with two distinct clusters. We observed significant correlations within the HIGH (r: M = 0.6011, SD = 0.2220, Mdn = 0.6429, p: M = 0.0307, SD = 0.1896, Mdn = 0.0222) and LOW (r: M = 0.6269, SD = 0.1697, Mdn = 0.6198, p: M = 0.0109, SD = 0.0322, Mdn = 0.0005) groups with a non-significant correlation between the HIGH and LOW groups (r: M = −0.0374, SD = 0.1849, Mdn = −0.0529, p: M = 0.4751, SD = 0.2611, Mdn = 0.4728). Figure 3B shows the distribution of Pearson’s “r” for within the HIGH, within the LOW, and the HIGH versus LOW groups.

Using KNN with one-holdout cross-validation, we were able to correctly predict every individuals’ membership in the HIGH and LOW groups.

Figure 4 shows the result of permutation importance on KNN. This test identified 27 channels whose contributions were important for the HIGH versus LOW groups’ classification. These channels were located in the frontal (Fp2, AF4, AFz, F1, Fz, F3, F5, F6), frontotemporal (FT7), central (C2, C6, Cz), centroparietal (CP1, CP3, CP4, CP5, CPz), parietal (P2, P3, P4, P5, P6, P7), parieto-occipital (PO3, PO4, POz), and occipital (O2) cortical regions. We also observed that 10 out these 27 channels had their importance > the average importance calculated by the permutation importance test (M = 0.0369, CI${}_{95.0\%}$ = [0.0227 0.0534]). These channels were in the frontal (AF4, F1, F6, Fz), centroparietal (CP3, CPz), parietal (P2, P3, P7), and occipital (O2) cortical regions.

Pairwise RSA with Pearson correlation similarity distance on these 10 channels (Figure 5A) resulted in improved distinction between the HIGH and LOW groups’ clusters. We observed significant correlations within the HIGH (r: M = 0.6971, SD = 0.2203, Mdn = 0.7416, p: M = 0.0184, SD = 0.0830, Mdn = 0.0003) and LOW (r: M = 0.7889, SD = 0.1180, Mdn = 0.8005, p: M = 0.0241, SD = 0.0313, Mdn = 0.0095) groups. Although the use of these channels increased the anti-correlation between the HIGH and LOW groups (r: M = −0.3393, SD = 0.1863, Mdn = −0.3504, p: M = 0.4139, SD = 0.2642, Mdn = 0.3551), it remained non-significant. Figure 5B shows the distribution of Pearson’s r for within the HIGH, within the LOW, and the HIGH versus LOW groups.

Interestingly, these 10 channels were sufficient for KNN with one-holdout cross-validation to correctly predict every individuals’ membership in the HIGH and LOW groups.

## 5. Discussion

The present study took a preliminary step toward the identification of an EEG-based resting-state digital marker for stress. In so doing, it utilized multi-channel, eye-closed, resting-state EEG recordings of 20 individuals who were chosen from a pool of 216 participants from the Max Planck Institute Leipzig Mind-Brain-Body Dataset [62]. These 20 individuals were those whose responses to “neuroticism” (FFM [71]), “worries” and “tension” (PSQ [72]), and the State-Trait Anxiety Inventory (STAI-G-X2 [73]) indicated a significantly higher and lower level of stress (10 individuals, per the HIGH and LOW groups), compared to all other participants.

The computed conditional entropy (CE) for EEG channels appeared to highlight the brain’s frontoparietal network [64]. This network, which is considered to form a domain-general network [81,82], plays a pivotal role in various brain functions [83,84]: from self-referential processing [65] and emotion [66] to social cognition [67], decision making [85], and working memory [86]. Therefore, it seems plausible to construe the observed sparsity of CEs pertinent to the components of this network in the HIGH group to highlight the potential impact of stress on their overall brain cognitive ability for handling various personal and social life events [11,87]. This interpretation gives further evidence from the finding that indicates the dampening effect of stress on higher level cognitive functions [31,33]. Moreover, the distribution of computed CEs appeared to identify the cortical components of the default mode network (DMN) [88,89,90]. This observation was interesting, considering the involvement of this network in the brain’s stress response [27,64,91,92].

The CE also appeared to be an effective EEG-based marker for stress. It not only captured the potential correlation among the members of the same stress group, but also showed a reliable specificity for distinguishing between individuals from the HIGH versus LOW groups. These observations presented further insight for the underlying relation between the entropic measures on the one hand and the brain signal variability on the other hand [56,57,58,59,60,61]. They also provided further evidence for recent findings [47] that showed that information from cortical entropy profiles could effectively predict diverse facets of human subjects’ cognitive ability.

Zhang et al. [30] also presented a high classification accuracy of pre- versus post-stress using the resting-state functional connectivity of healthy individuals. The present results extended their study to the case of EEG, thereby providing an alternative to the limited applicability of fMRI in more naturalistic settings. They also showed the potential of the CE as a reliable marker for detecting the effect of stress on brain function in the absence of any explicit stressors.

From a broader perspective, our results posit the use of the CE as a potential EEG-based diagnostic marker of the stress. For instance, the CE can help identify the individuals that are at higher risk of stress-related neuropsychological disorders. Subsequently, it may also prove useful for tracking the effect of stress-related treatments on these patients through quantification of the changes in the pattern of cortical information processing in comparison with healthy individuals.

## 6. Limitations and Future Direction

The present study did not include female participants due to their limited number in the final selected HIGH and LOW groups’ members. Lighthall et al. [93] and Seo et al. [94] showed the differential effect of participants’ gender on the brain response to stress. This necessitates further investigation of our findings in settings in which both male and female (as well as more gender-diverse individuals) are included. Similarly, the present study excluded older adults since their limited number (i.e., one individual in each of the LOW and HIGH stress groups) was not sufficient to verify whether the observed results were due to stress or the effect of aging on reduced complexity and information processing capacity of the brain [95,96]. As a result, it is crucial to consider the sample population that was comprised of older people, as well as adolescents [2,91] to draw more informed conclusions on the change in cortical information processing in response to stress.

Our approach to selecting the individuals for the present study primarily focused on those participants whose responses formed the two extremes of the stress spectrum, i.e., the HIGH and LOW groups. This meant that we were not able to verify whether the differences in the CEs were due to the significantly different mindsets of the HIGH versus LOW groups (as far as their subjective responses to the questionnaires were concerned) or it rather captured a substantial change in the brain functions whose gradual effect could be traced along the stress-effect spectrum. Considering the findings that identified the individuals’ subjective ratings to best predict their distress across a variety of self-reported measures [97], future research can broaden the scope of the present study to the case in which individuals with broader stress responses are included. This, in turn, can allow for more informed conclusions on the generalizability of the observed differences between the HIGH and LOW stress groups to a more inclusive scenario in which a wider range of stress responses is considered. Such studies can benefit from the application of advanced clustering algorithms such as the successive geometrical transformations model (SGTM) neural-like structure [98,99,100] that show promising results for medical images’ pre-processing, classification, and recognition.

Pairwise computation and analysis of information-theoretic measures in a system with multiple interacting processes are subject to several shortcomings (Wibral et al. [55], pp. 24–25). These observations demonstrate the need for surpassing the bivariate formulation (i.e., pairwise correspondences, as presented in the present study) of such entropic measures as the conditional entropy, thereby studying the brain information processing in light of all its interacting components. Although there is no constraint that would limit the extension of the bivariate case to the case of multivariate analysis of the brain information processing, the exponential growth in computational cost associated with such multivariate analyses confines their application to more practical and affordable scenarios such as the study of binary circuits [101] and neural cultures [102]. On the other hand, the on-going efforts and progress for devising efficient tools for such multivariate analyses of brain function [103] bring hope for their possibilities in the future.

## Figures and Tables

**Figure 1 entropy-23-00286-f001:**
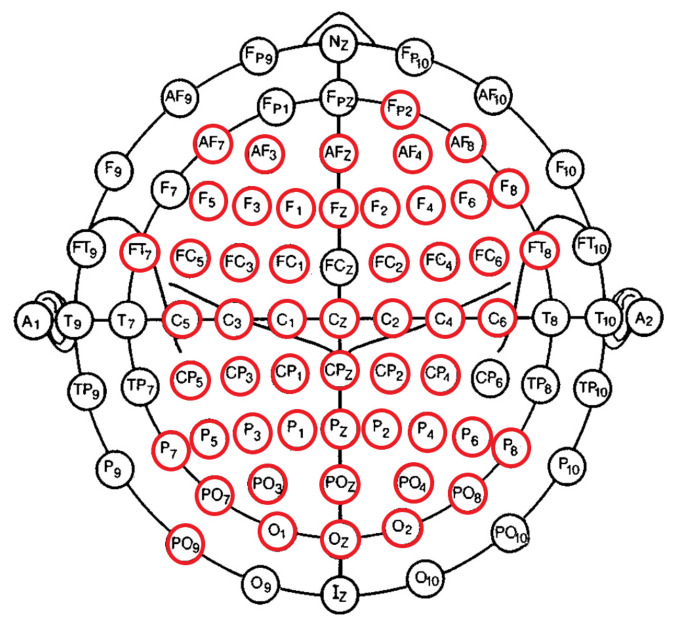
Fifty-three EEG channels (circled in red) that were commonly available in all participants’ pre-processed EEG recordings. These channels were FP2, AF7, AF3, AFZ, AF4, AF8, C5, C3, C1, Cz, C2, C4, C6, CP5, CP3, CP1, CPZ, CP2, CP4, F5, F3, FZ, F1, F2, F4, F6, F8, FT7, FC5, FC3, FC1, FC2, FC4, FC6, FT8, P7, P5, P3, P1, PZ, P2, P4, P6, P8, PO9, PO7, POZ, PO3, PO4, PO8, and O1, OZ, O2.

**Figure 2 entropy-23-00286-f002:**
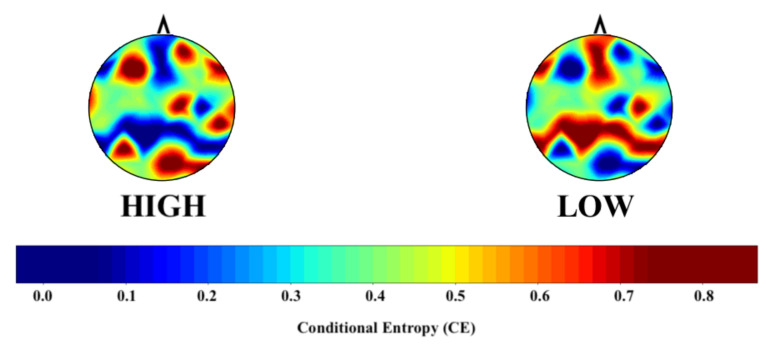
Grand average topographic maps of the HIGH and LOW groups’ conditional entropy (CE) values. This figure highlights the activation of the default mode network’s (DMN) frontoparietal component in the LOW group, which appears impaired in the LOW group. The CE values are scaled using the “*StandardScaler()*” function from Python scikit-learn [78].

**Figure 3 entropy-23-00286-f003:**
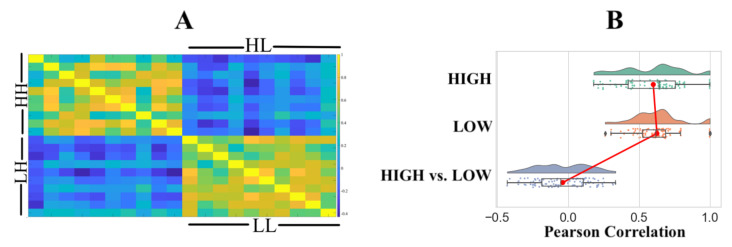
(**A**) Representational similarity analysis (RSA) using Pearson correlation. HH and LL refer to within-group HIGH and within LOW comparison. RSA comparison between the HIGH and LOW groups is identified by HL and LH. (**B**) Distribution of Pearson’s “r” within the HIGH, within the LOW, and between the HIGH and LOW groups.

**Figure 4 entropy-23-00286-f004:**
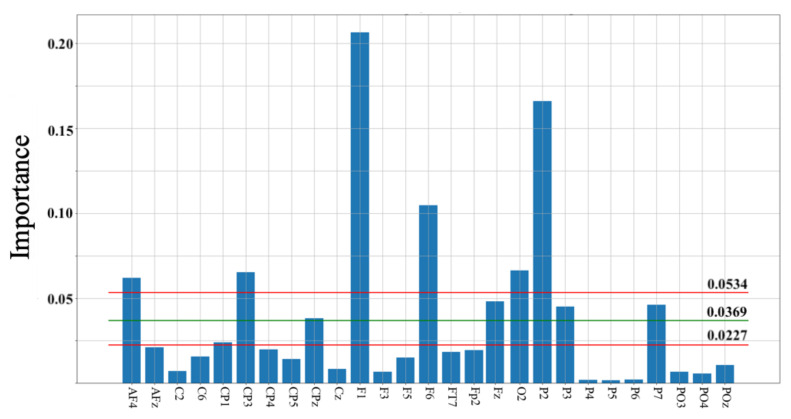
Permutation importance on KNN identified 27 channels. These channels were: AF4, AFz, C2, C6, CP1, CP3, CP4, CP5, CPz, Cz, F1, F3, F5, F6, FT7, Fp2, Fz, O2, P2, P3, P4, P5, P6, P7, and PO3, PO4, POz. Green and red horizontal lines mark the mean and 95.0% confidence interval (CI) for permutation importance values (M = 0.0369, CI${}_{95.0\%}$ = [0.0227 0.0534]). In this figure, channels whose importance are > average are in the frontal (AF4, F1, F6, Fz), centroparietal (CP3, CPz), parietal (P2, P3, P7), and occipital (O2) cortical regions.

**Figure 5 entropy-23-00286-f005:**
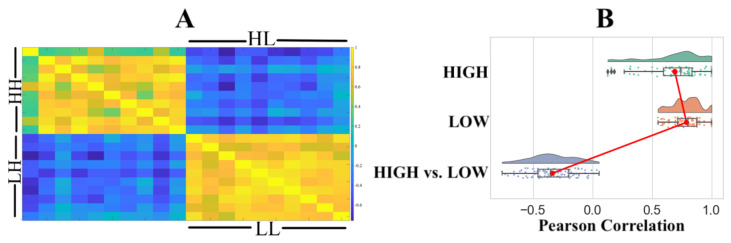
Representational similarity analysis (RSA) based on channels whose permutation importance was above average (Figure 4, M = 0.0369). (**A**) HH and LL refer to within the HIGH and within the LOW groups’ comparison. RSA comparison between the HIGH and LOW groups is identified by HL and LH. (**B**) Distribution of Pearson’s r within the HIGH, within the LOW, and between the HIGH and LOW groups.

## Data Availability

Data related to the present study is available at http://fcon_1000.projects.nitrc.org/indi/retro/MPI.html, accessed on 25 February 2021 and [62].

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
