# Peer review of "Conditional Entropy: A Potential Digital Marker for Stress"

_entropy, 2021, doi:10.3390/e23030286_

Round 1

Reviewer 1 Report

Paper deals with important task in medicine. Authors proposed an interesting approach for identification digital marker of stress using resting-state multi-channel electroencephalography (EEG) recording/ Authors used conditional entropy (CE) as a tool for that.

Paper has great practical value.

It has a logical structure, all necessary sections. Paper is technically sound. Experimental section is good.

The proposed approach are logical, results are clear.

Suggestions:

  1. It would be good to add point-by-point the main contributions in the end of the Introduction section
  2. It would be good to add the reminder of this paper
  3. Related works section should be extended using SGTM neural-like structure as an effective and fast tool for clustering
  4. Authors should provide an information about optimal number of clusters used in this study. How it was chosen?
  5. A lot of references are outdated. Please fix it using 3-5 years old papers in high-impact journals

Author Response

First and foremost, we are thankful for the reviewer’s time and kind consideration to review the present manuscript. Reviewer’s comments substantially improved the quality of the present study and its presentation.

In what follows, point-by-point responses to reviewer’s comments and concerns are provided.

Sincerely,

Reviewer 1

Reviewer’s Comment: It would be good to add point-by-point the main contributions in the end of the Introduction section

Author’s Response: We added a paragraph, highlighting the main contributions of the present study in Section 1. Introduction, lines 92-97, in the current version of manuscript. It reads as follow.

The contribution of the present study is twofold. First, we extend the previous research that were based on brain stress-response to psychological and physical stressors to the case of resting-state brain activity. In so doing, we show that CE highlights the effect of stress on the brain's frontoparietal network [64] that plays a pivotal role in self-referential processing [65], emotion [66], and social cognition [67]. Second, we show that CE presents an effective EEG-based resting-state digital marker for quantification of the brain distributed response to stress.

Reviewer’s Comment: It would be good to add the reminder of this paper

Author’s Response: We added a paragraph, summarizing the structure of the manuscript at the end of Section Introduction, Lines 98-101, as follows.

The remainder of this article is organized as follows. Section 2 provides details on dataset, resting-state EEG recordings, and their pre-processing. Section 3 summarizes the analysis steps adapted in the present study. Section 4 presents the results. Discussion along with the limitations and future direction of the present study are presented in Sections 5 and 6.

Reviewer’s Comment: Related works section should be extended using SGTM neural-like structure as an effective and fast tool for clustering

Author’s Response: We included SGTM in Section 6. Limitations and Future Direction, lines 306-309, in the current version of manuscript as follows.

Such studies can benefit from application of advanced clustering algorithms such successive geometrical transformations model (SGTM) neural-like structure [102-104] that show promising results for for medical images' pre-processing, classification, and recognition.

We would like to ask the reviewer to please let us know if we have missed important references for SGTM so we can include them.

Reviewer’s Comment: Authors should provide an information about optimal number of clusters used in this study. How it was chosen?

Author’s Response: We did not use any clustering algorithm in our study. There were two groups of participants (i.e., groups comprising individuals with high and low stress responses) whose assignments to their respective groups were done based on analysis of their responses to "neuroticism" from the big-five of personality [73], "worries" and "tension" from perceived stress questionnaire (PSQ) [74], and the state-trait anxiety inventory (STAI-G-X2) [75]. We explained this process in Section 2.4. Choice of Individuals with High and Low Stress Responses, lines 142-169 and Appendix A. Descriptive Statistics of Participants responses to each of FFMs neuroticism, PSQs worries and tension, and STAI-G-X2's state-trait anxiety, lines 331-332 along with Table A1, page 10.

Reviewer’s Comment: A lot of references are outdated. Please fix it using 3-5 years old papers in high-impact journals

Author’s Response: We thank the reviewer for highlighting this shortcoming of our manuscript. We revised the literature reported in our study and substantially improved its coverage of the recent findings and advances in identification of stress markers. In particular, we included a number of recent findings that specifically made use of various entropic measures for this purpose. Subsequently, we revised and rewrote the manuscript’s Abstract. Although we think that we cited the more recent literature, we would be thankful if the reviewer could point at particular references that we might have missed.

Reviewer 2 Report

This work introduces the conditional entropy for detection of stress by means of the analysis of electroencephalographic recordings. I have some concerns about this manuscript:

1.- Lines 30-31: “…, there is a paucity of research on utilization of the brain activity for identification of effective stress markers.” As a researcher in this area, I totally disagree with the author of this manuscript. During the last years, a vast number of studies have focused on the study of EEG signals for stress detection. Many of them have applied entropy metrics for that purpose. More precisely, the conditional entropy has already been applied for recognition of stress with EEG signals. In this sense, the main idea of this manuscript is not a novelty in the scientific literature. I would strongly recommend the author making a thorough revision of the studies published during the last years in this research field and include it in the introduction section as a revision of the state-of-the-art, and make sure this work is actually providing novel contributions.

2.- Typically, studies for recognition of emotions such as stress with EEG and other physiological recordings make use of images, sounds, or video clips as stimuli. Nevertheless, in this work the EEG signals analyzed were registered during intervals of eyes open and closed of individuals in a resting state, without using any kind of stimulus for stress elicitation. Why have those signals been selected for stress detection? Does that method present any advantage with respect to the audiovisual stimulation?

3.- A more in-depth mathematical definition of the conditional entropy should be given for the interest of non-expert readers. According to eq. 2, does conditional entropy of one EEG channel depend on the rest of channels?

4.- In figure 2, how could it be explained that CE values of the EEG channels represented are either the maximum or the minimum in the whole scale? In other words, channels are either colored with dark blue (minimum CE) or dark red (maximum CE). What about the values in the middle of the scale?

Author Response

First and foremost, we are thankful for the reviewer’s time and kind consideration to review the present manuscript. Reviewer’s comments substantially improved the quality of the present study and its presentation.

In what follows, point-by-point responses to reviewer’s comments and concerns are provided.

Sincerely,

Reviewer 2

Reviewer’s Comment: Lines 30-31: “…, there is a paucity of research on utilization of the brain activity for identification of effective stress markers.” As a researcher in this area, I totally disagree with the author of this manuscript. During the last years, a vast number of studies have focused on the study of EEG signals for stress detection. Many of them have applied entropy metrics for that purpose. More precisely, the conditional entropy has already been applied for recognition of stress with EEG signals. In this sense, the main idea of this manuscript is not a novelty in the scientific literature. I would strongly recommend the author making a thorough revision of the studies published during the last years in this research field and include it in the introduction section as a revision of the state-of-the-art, and make sure this work is actually providing novel contributions.

Author’s Response: We thank the reviewer for highlighting this shortcoming of our manuscript. We revised the literature reported in our study and substantially improved its coverage of the recent findings and advances in identification of stress markers. In particular, we included a number of recent findings that specifically made use of various entropic measures for this purpose. Subsequently, we revised and rewrote the manuscript’s Abstract. Although we think that we cited the entropic-related literature, we would be thankful if the reviewer could point at particular references that we might have missed.

With regard to the study contributions, we added the following to Section 1. Introduction, lines 92-97, in the current version of manuscript.

The contribution of the present study is twofold. First, we extend the previous research that were based on brain stress-response to psychological and physical stressors to the case of resting-state brain activity. In so doing, we show that CE highlights the effect of stress on the brain's frontoparietal network [64] that plays a pivotal role in self-referential processing [65], emotion [66], and social cognition [67]. Second, we show that CE presents an effective EEG-based resting-state digital marker for quantification of the brain distributed response to stress.

Reviewer’s Comment: Typically, studies for recognition of emotions such as stress with EEG and other physiological recordings make use of images, sounds, or video clips as stimuli. Nevertheless, in this work the EEG signals analyzed were registered during intervals of eyes open and closed of individuals in a resting state, without using any kind of stimulus for stress elicitation. Why have those signals been selected for stress detection? Does that method present any advantage with respect to the audiovisual stimulation?

Author’s Response: Van Oort et al. [27] observed that the use of different types of experimentally induced psychological and physical stressors by most of previous studies could potentially yield differential impacts on the brain response to stress. They further asserted that such variations must be dissociated from more (potentially) general patterns. The present study took a step toward addressing this issue by focusing on EEG-based resting-state digital marker of stress. We included this information in Section 1. Introduction, lines 77-91, in the current version of manuscript as follows.

The overview of the literature on the use of brain activity for identification of a stress-marker show a substantial progress in this direction. In particular, the use of entropic measures [50,51, 54] is very appealing, considering (1) their suitability for capturing both linear and non-linear characteristics of EEG time series [55] (2) their direct correspondence with the brain signal variability [47, 56-61]. At the same time, these findings rely on external stimuli to induce the brain stress-response. In this respect, Van Oort et al. [27] observed that the use of different types of experimentally induced psychological and physical stressors by most of previous studies could potentially yield differential impacts on the brain response to stress. They further asserted that such variations must be dissociated from more (potentially) general patterns.

        The present study takes a step toward addressing this issue by introducing conditional entropy (CE) as a potential EEG-based resting-state digital marker of stress. For this purpose, we use the resting-state EEG recordings of human subjects from Max Planck Institute Leipzig Mind-Brain-Body Dataset [62] The participants in this study did not perform any stress-related task and only completed the multidimensional mood state (MDBF) questionnaire [63] prior to their resting-state EEG recordings (5-point Likert scale, from 1 (not at all) to 5 (very much) [62]).

Reviewer’s Comment: A more in-depth mathematical definition of the conditional entropy should be given for the interest of non-expert readers. According to eq. 2, does conditional entropy of one EEG channel depend on the rest of channels?

Author’s Response: We extended Section 2.6. Conditional Entropy (CE) Computation, lines 179-193, in the current version of manuscript to provide further explanation (along with additional equations) about CE.

With regard to the part of reviewer’s comment According to eq. 2, does conditional entropy of one EEG channel depend on the rest of channels?”: Conditional entropy of a random variable X, given another random variable Y (in our case two EEG channels) quantifies the amount uncertainty that remains about X when Y is known/observed. We provided this information in Section 2.6. Conditional Entropy (CE) Computation, lines 182-185, in the current version of manuscript as follows.

CE quantifies the amount uncertainty that remains about X (i.e., its information content) when Y is known/observed. Equivalently, CE of X given Y can be computed based on the relation between the amount of information that is shared by X and Y (i.e., their mutual information (MI)) and X's entropy [74]:(Remarks: This is followed by newly added equation (2) in the current version of manuscript.)

As such,  (i.e., equation 3 in the current version of manuscript) quantifies average information content in EEG channel  after its all pairwise conditional entropies with respect to the rest of the EEG channels (i.e., all information in  that could be explained by observing these other channels) are accounted for. We added this information in Section 2.6. Conditional Entropy (CE) Computation, lines 190-193, in the current version of manuscript as follows.

Therefore,  in equation 3 quantifies the average information content in EEG channel  after all its pairwise conditional entropies with respect to the remaining EEG channels (i.e., all information in  that could be explained by observing  channels) are accounted for.

We also realized the necessity to acknowledge the limitations of present and other reviewed findings that make use of entropic measures. Precisely, these results are based on all-pair (and therefore bivariate in nature) formulation of these measures. Although their extension to multivariate scenario is theoretically possible, the exponential growth in computation cost associated with multivariate studies makes them (for the time being) intractable, some recent advances bring hope for their availability in the future. We included this discussion in Section 6. Limitations and Future Direction, line 310-320, in the current version of manuscript. It reads as follows.

Pairwise computation and analysis of information-theoretic measures in a system with multiple interacting processes is subject to several shortcomings (Wibral et al [55], p. 24-25). These observations demonstrate the need for surpassing the bivariate formulation (i.e., pairwise correspondences, as presented in the present study) of such entropic measures as conditional entropy, thereby studying the brain information processing in light of all its interacting components. Although there is no constraint that would limit the extension of bivariate to case of multivariate analysis of the brain information processing, the exponential growth in computational cost associated with such multivariate analyses confine their application to more practical and affordable scenarios such study of binary circuits [105] and neural cultures [106]. On other hand, the on-going efforts and progress for devising efficient tools for such multivariate analyses of the brain function [107] brings hope for their possibilities in the future.

Reviewer’s Comment: In figure 2, how could it be explained that CE values of the EEG channels represented are either the maximum or the minimum in the whole scale? In other words, channels are either colored with dark blue (minimum CE) or dark red (maximum CE). What about the values in the middle of the scale?

Author’s Response: We thank the reviewer for this important comment. Please note that the values in this figure are not solely based on two binary color-coded values (i.e., dark-blue and dark-red) but a range that falls between these two extremes.

Although it is possible to compute entropy measures using parametric approaches, it would inevitably require an assumption about shape of a probability density (e.g., data is derived from a Gaussian distribution). To avoid imposing such constraint/assumption on our data, we opted for non-parametric estimation of CE. When using such non-parametric estimators, a crucial step is to verify whether the observed non-zero estimates are due to such issues as limited data and/or non-stationarity that could be present in time series. Therefore, it is important to ensure that the observed non-zero entropic values are indeed significant through application of permutation test on surrogate data during their computation [55]. We achieved this objective through “null distribution and statistical significance” functionality that is available in JIDT [81] which we used in our study. Referring to Figure 2 in the current version of manuscript, the regions that are depicted in dark-blue correspond to the channels whose computed CE did not pass the aforementioned null distribution based on surrogate data. We added this information in Section 3. Analysis, lines 207-217 as follows.

We computed CE values using Python version of JIDT [81]. With regard to CE computation, there is a crucial point that derves further explanation. Although it is possible to compute entropic measures using parametric approaches, it would inevitably require an assumption about shape of a probability density (e.g., data is derived from a Gaussian distribution)[82]. To avoid imposing such constraint/assumption on our data, we opted for non-parametric estimation of CE [81]. When using such non-parametric estimators, a crucial step is to verify whether the observed non-zero estimates are due to such issues as limited data and/or non-stationarity that could be present in time series. Therefore, it is important to ensure that the observed non-zero entropic values are indeed significant through application of permutation test on surrogate data during their computation [55]. We achieved this objective through "null distribution and statistical significance" functionality that is available in JIDT [81] using 100 rounds of permutation test (computeSignificance(100) in JIDT).

Round 2

Reviewer 2 Report

With this new version, authors have improved the background of the manuscript. Most of the suggestions in first review step have been followed. However, I still have a main concern with my last question.

When talking about the representations of CE values in Figure 2, authors talk about the possibility of making a parametric or non-parametric calculation of CE. It should be better explained, since CE computation should be done according to equations 1 to 3. I cannot see the relationship between parametric/non-parametric and those equations...

In any case, the question was to justify why electrodes presented either the minimum or the maximum value of the scale. Authors say that the whole scale is used, however it is not exactly that manner. Indeed, the intermediate values are only used in the points BETWEEN electrodes, but each electrode still has either the maximum or the minimum values of CE. Please, revise this crucial point, since it is hard to understand those strange results...

On the other hand, please revise the whole text for correction of several English mistakes and typos, especially in the new parts of the manuscript.

Author Response

First and foremost, we are thankful for the reviewer’s time and kind consideration to review the present manuscript. Reviewer’s comments substantially improved the quality of the present study and its presentation.

In what follows, point-by-point responses to reviewer’s comments and concerns are provided.

Sincerely,

Reviewer 2

Reviewer’s Comment: When talking about the representations of CE values in Figure 2, authors talk about the possibility of making a parametric or non-parametric calculation of CE. It should be better explained, since CE computation should be done according to equations 1 to 3. I cannot see the relationship between parametric/non-parametric and those equations...

Author’s Response: We extended this explanation and more clearly explained the relation between parametric approaches and equations 1 through 3. The modified content in the current version of the manuscript (Section 3. Analysis, lines 207-222) reads as follows.

We computed CE values using Python version of JIDT [79]. With regard to CE computation, there is a crucial point that deserves further explanation. While computing CE, it is possible to opt for parametric approaches, thereby bypassing estimation of the joint and conditional probabilities in equation (1) and instead utilizing available analytical forms. For instance, assuming X and Y were Gaussian random variables, then one could make use of (please see PDF version of this response for the equation [74, p. 244], , and (please see PDF version of this response for the equation)  [74, p. 252] (where (please see PDF version of this response for variance symbol) and (please see PDF version of this response for variance symbol) are X and Y respective variance and (please see PDF version of this response for correlation symboldenotes their correlation) to compute their CE. However, such an approach would inevitably require an assumption about shape of underlying probability densities of the random variables under consideration (e.g., Gaussianity of X and Y in above example) [80]. To avoid imposing such constraint/assumption on our data, we opted for non-parametric estimation of CE [79]. When using such non-parametric estimators, a crucial step is to verify whether the observed non-zero estimates are due to such issues as limited data and/or non-stationarity that could be present in time series. Therefore, it is important to ensure that the observed non-zero entropy values are indeed significant through application of permutation test on surrogate data during their computation [55]. We achieved this objective through "null distribution and statistical significance" functionality that is available in JIDT [79] using 100 rounds of permutation test (computeSignificance(100) in JIDT).

Reviewer’s Comment: In any case, the question was to justify why electrodes presented either the minimum or the maximum value of the scale. Authors say that the whole scale is used, however it is not exactly that manner. Indeed, the intermediate values are only used in the points BETWEEN electrodes, but each electrode still has either the maximum or the minimum values of CE. Please, revise this crucial point, since it is hard to understand those strange results...

Author’s Response: In the previous version of this figure, we only included those channels whose CE values with respect to all other channels passed the permutation test (i.e., Section 3. Analysis, lines 211-217 in the previous version of manuscript and lines 216-222 in the current version) and subsequently scaled them using “StandardScaler” from Python scikit learn. To address the reviewer’s comment, we regenerate this figure for the current version (page 6). Please note that the values in this figure are scaled for comparability. We added this note to the figure’s caption as follows.

 “Grand average topographic maps of HIGH and LOW groups' CE values. This figure highlights activation of DMN's frontoparietal component in LOW group that appears impaired in LOW group. The CE values are scaled using “StandardScaler()” function from Python scikit-learn [78].

Reviewer’s Comment: On the other hand, please revise the whole text for correction of several English mistakes and typos, especially in the new parts of the manuscript

Author’s Response: We thank the reviewer for a thorough review of the manuscript. We read the manuscript carefully and corrected for typos and also checked for its grammar.
